# Susceptibility of β-Thalassemia Heterozygotes to COVID-19

**DOI:** 10.3390/jcm10163645

**Published:** 2021-08-18

**Authors:** Sotirios Sotiriou, Athina A. Samara, Dimitra Vamvakopoulou, Konstantinos-Odysseas Vamvakopoulos, Andreas Sidiropoulos, Nikolaos Vamvakopoulos, Michel B. Janho, Konstantinos I. Gourgoulianis, Styllianos Boutlas

**Affiliations:** 1Department of Embryology, Faculty of Medicine, School of Health Sciences, University of Thessaly, 41110 Larissa, Greece; sotiriousoti@yahoo.gr (S.S.); kostantinos753@gmail.com (K.-O.V.); Micheljanho@live.co.uk (M.B.J.); 21st Neonatal Intensive Care Unit “Agia Sophia” Children’s Hospital, 11527 Athens, Greece; gina_dimitra@hotmail.com; 3Cardiology Department, Papageorgiou General Hospital, 56403 Thessaloniki, Greece; achsid@gmail.com; 4Department of Biology, Faculty of Medicine, University of Thessaly, 41110 Larissa, Greece; nvamvak@yahoo.com; 5Department of Respiratory Medicine, Faculty of Medicine, University of Thessaly, 41110 Larissa, Greece; kgourg@med.uth.gr (K.I.G.); sboutlas@gmail.com (S.B.)

**Keywords:** COVID-19, β-thalassemia, risk, coronavirus

## Abstract

Background: β-Thalassemia is the most prevalent single gene blood disorder, while the assessment of its susceptibility to coronavirus disease 2019 (COVID-19) warrants it a pressing biomedical priority. Methods: We studied 255 positive COVID-19 participants unvaccinated against severe acute respiratory syndrome–coronavirus 2 (SARS-CoV-2), consecutively recruited during the last trimester of 2020. Patient characteristics including age, sex, current smoking status, atrial fibrillation, chronic respiratory disease, coronary disease, diabetes, neoplasia, hyperlipidemia, hypertension, and β-thalassemia heterozygosity were assessed for COVID-19 severity, length of hospitalization, intensive care unit (ICU) admission and mortality from COVID-19. Results: We assessed patient characteristics associated with clinical symptoms, ICU admission, and mortality from COVID-19. In multivariate analysis, severe-critical COVID-19 was strongly associated with male sex (*p* = 0.023), increased age (*p* < 0.001), and β-thalassemia heterozygosity (*p* = 0.002, OR = 2.89). Regarding the requirement for ICU care, in multivariate analysis there was a statistically significant association with hypertension (*p* = 0.001, OR = 5.12), while β-thalassemia heterozygosity had no effect (*p* = 0.508, OR = 1.33). Mortality was linked to male sex (*p* = 0.036, OR = 2.09), increased age (*p* < 0.001) and β-thalassemia heterozygosity (*p* = 0.010, OR = 2.79) in multivariate analysis. It is worth noting that hyperlipidemia reduced mortality from COVID-19 (*p* = 0.008, OR = 0.38). No statistically significant association of current smoking status with patient characteristics studied was observed. Conclusions: Our pilot observations indicate enhanced mortality of β-thalassemia heterozygotes from COVID-19.

## 1. Introduction

Identifying medical conditions with a high or potentially deadly impact on the disease caused by severe acute respiratory syndrome coronavirus 2 (SARS-CoV-2), is a critical initial step towards containment of associated morbidity and mortality risks. Given that viral stress from SARS-CoV-2 elicits anabolic responses supported by increasing blood pressure to meet enhanced oxygen needs of vital organs and organ systems, hypoxemia is rendered a high-risk medical condition [1,2]. As the most common blood disorder affecting approximately one third of the global population, anemia presents a low tolerance to hypoxemia and may have either acquired polysystemic or inherited poly- or monogenic background [3]. Monogenic anemia—which is caused by abnormal hemoglobin—is a rather prevalent medical disorder with 270 million carriers worldwide [4,5,6]. β-Thalassemia is the most common inherited single gene disorder in the world. Approximately one-third of all hemoglobinopathies and/or nearly 1.5% of the global population carry the β-thalassemia trait [7]. In this context, β-thalassemia heterozygosity is a strong candidate condition for assessing an individual’s susceptibility to COVID-19.

In the present study, we aimed to compare the effect of age, sex, complex co-morbidities, and β-thalassemia status on clinical outcomes. It was determined that β-thalassemia heterozygotes were more likely to develop severe and critical COVID-19 (*p* = 0.002, OR = 2.89) or die from the disease (*p* = 0.01, OR = 2.79); however, β-thalassemia heterozygotes were not likely to be admitted to the hospital’s intensive care unit (ICU) (*p* = 0.508, OR = 1.33). These findings suggest that β-thalassemia heterozygotes present increased morbidity and mortality related to COVID-19, and therefore supports the urgent need for their recognition as a high-risk group to facilitate early identification, consultation, and intervention. 

## 2. Material and Methods

### 2.1. Patients

Our study population included 255 participants who were not vaccinated against COVID-19 and had a positive SARS-CoV-2 Real-Time Polymerase Chain Reaction (RT-PCR) molecular test. All study participants provided consent to participate in the study. The mean age of participants was 61.56 (±16.597) years, ranging from 20 to 92 years of age. Participants were consecutively recruited through their admittance to the emergency department (ER) of a tertiary referral center in central Greece (Larisa University Hospital), between 1 October and 31 December 2020. Of those patients, 153 (60%) were male and 102 (40%) were female. Current smoking status of participants was recorded in only 79 (31%) study participants. 

### 2.2. Patient Symptoms and Study Design

Detailed clinical characteristics of SARS-CoV-2 (dominant variant 20B/GR clade), as well as the corresponding treatment protocols, are available online [8].Participants were examined with chest X ray or chest CT and categorized into groups based on ascending severity of viral symptoms as follows: non-hospitalized, either asymptomatic or presenting with mild illness (26.7%, 68 patients) and hospitalized with either moderate (44.3%, 113 patients) or severe to critical illness 74 (29%, 74 patients) according to their dependence on oxygen support (Table 1). A clinical and demographic database was created based on both participants’ reported and clinically re-assessed medical history of confirmed COVID-19 positive patients. Statins were prescribed to all our hyperlipidemic and in association with anti-hypertensive treatment to a small number of hypertensive study participants. The clinical course of non-hospitalized participants was followed by conducting telephone interviews. Overall, 53 (20.8%) study participants were admitted to the ICU (Table 2) and 70 (27.5%) study participants died (Table 3). COVID-19 infection was the single common official cause of death for our study participants that was registered in hospital archives.

The present study was designed to assess the association of clinical and demographic characteristics to the outcome of study participants. In this context, we recorded patients’ age, sex, current smoking status, and history of major comorbidities (such as atrial fibrillation, chronic respiratory disease, coronary disease, diabetes, neoplasia, hyperlipidemia, hypertension, and β-thalassemia heterozygosity), in relation to severity of clinical symptoms, time of hospitalization, ICU admission and mortality due to COVID-19 (Table 1, Table 2 and Table 3). Other than being COVID-19 positive, our β-thalassemia trait carriers with Ht 32 to 39, were free from additional rare hemoglobin variant combinations or notable comorbidities requiring hypertensive and hyperlipidemic medication. The study was conducted in accordance with the Research and Ethical Committee guidelines of the Larisa University Hospital.

### 2.3. Statistical Analysis

Statistical analysis was performed using SPSS v.25 (IBM, Chicago, IL, USA). Qualitative variables are presented as absolute (N) and relative (%) frequencies, and quantitative variables are presented as means with standard deviation (SD). Both the Mann–Whitney U test and Kruskal–Wallis test were used for continuous variables where data did not follow a normal distribution. Normal distribution was tested using the Shapiro–Wilk normality test. Qualitative variables were analyzed using Chi-square test or Fisher’s exact test. In addition, multivariate analysis was performed using binary and ordinal logistic regression. The significance level was set at 5% (0.05). 

#### Sample Estimation

Considering an estimated prevalence of 8% in our entire study population, a precision of ±3.5% and a 95% confidence interval (CI), the minimum sample size required was calculated by a precision analysis using Epi Info 7 [9]. It was determined to be 231 patients.

## 3. Results

Association of β-thalassemia heterozygosity with severe and critical COVID-19 symptoms

Considering the clinical spectrum of COVID-19 as a primary outcome, patients were categorized into three groups (asymptomatic and mild/ moderate/ severe and critical). No difference in chest X ray or CT scan was observed among study participants. In univariate analysis, sex (*p* = 0.047), age (*p* < 0.001), atrial fibrillation (*p* = 0.022), coronary disease (*p* = 0.041), hyperlipidemia (*p* = 0.014), hypertension (*p* < 0.001), and being heterozygous for thalassemia (*p* = 0.004) were associated with severe COVID-19 symptoms (Table 1). In multivariate analysis, male sex (*p* = 0.023), increased age (*p* < 0.001), and being heterozygous for thalassemia (*p* = 0.002) were identified as independent risk factors for severe and critical clinical COVID-19 symptoms. Specifically, males had a 1.81 times (95% CI, 1.09 to 3.01) increased possibility for severe or critical clinical symptoms; increased age was associated with increased odds of severe and clinical symptoms with OR = 1.06 (95% CI, 1.04 to 1.08). A finding of great interest is that patients who were heterozygous for thalassemia were 2.89 times (95% CI, 1.49 to 5.62) more likely to have severe and critical clinical symptoms of COVID-19 (Figure 1).

### 3.1. Association of β-Thalassemia Heterozygotes with Mortality Due to COVID-19

Regarding mortality associated with COVID-19 infection, in univariate analysis sex (*p* = 0.022), age (*p* < 0.001), atrial fibrillation (*p* = 0.002), respiratory disease (*p* = 0.027), coronary disease (*p* = 0.027), hypertension (*p* < 0.001), and being heterozygous for thalassemia (*p* = 0.005) were associated with mortality (Table 2). In logistic regression analysis, male patients had a 2.09 times (95% CI, 1.05 to 4.18) greater possibility of dying and patients with increased age were 1.06 times (95% CI, 1.03 to 1.09) more likely to die. It is worth noting that hyperlipidemia plays a beneficial role in COVID-19 mortality, as the odds ratio of mortality in patients with hyperlipidemia is 0.65 (95% CI 0.37–1.15). It should be highlighted that patient who are heterozygous for thalassemia have a 2.79 times (95% CI, 1.28 to 6.09) greater possibility of dying than other patients (Figure 2).

### 3.2. Admission of COVID-19 Infected β-Thalassemia Heterozygotes to the ICU

Regarding the requirement for ICU care, it was found through univariate analysis that age (*p* = 0.03), respiratory disease (*p* = 0.043), coronary disease (*p* = 0.029) and hypertension (*p* < 0.001) were associated with ICU admission (Table 3). Through logistic regression analysis, patients with hypertension had 5.12 times (95% CI, 2.04 to 12.87) greater risk of requiring ICU care than patients without hypertension. On the contrary, hyperlipidemia was identified as a protective factor against ICU admission, with OR = 0.44 (95% CI, 0.21 to 0.94). Furthermore, in relation to the requirement for ICU care, being heterozygous for thalassemia had no effect on the possibility of admission to the ICU (*p* = 0.505).

### 3.3. Length of Hospitalization until Death

When comparing the median length of hospitalization (days) between patients being heterozygous for thalassemia and non-carriers, a statistically significant difference was observed (*p* = 0.046) (Figure 3). More specifically, the median duration of hospitalization among carriers and non-carriers was 12 and 17.5 days, respectively. 

### 3.4. Length of Hospitalization among Patients Who Survived

Regarding days of hospitalization among patients that survived COVID-19, the median duration was eight days for patients that were heterozygous for thalassemia and six days for non-carriers (*p* = 0.014) (Figure 4).

## 4. Discussion

In this pilot study, we aimed to evaluate the impact of COVID-19 on β-thalassemia heterozygotes. We assessed the association of age, sex, common co-morbidities, and β-thalassemia heterozygosity to clinical outcomes of participants from central Greece who were not vaccinated against SARS-CoV-2 and tested positive for COVID-19 during the last trimester of 2020. Our findings support earlier observations that male sex and older age are associated with poorer outcomes [9,10,11,12,13] and suggest that hyperlipidemia reduced participant mortality due to COVID-19 (*p* = 0.08, OR = 0.38), while β-thalassemia heterozygosity enhanced (*p* = 0.010, OR = 2.79) participant mortality due to COVID-19 (Table 3, Figure 2).

While hyperlipidemia and obesity are common co-morbidities with negative impacts on most associated pathologies [14], in the present study patients with hyperlipidemia were clearly protected from mortality due to COVID-19, with a 60% lesser probability of death compared to other patients (OR = 0.38). Known to reduce COVID-19 mortality [15,16,17], statins that were prescribed to hyperlipidemic patients may account for patients’ protection against COVID-19 associated death. In addition to their primary lipid-lowering effect, statins appear to regulate vasodilation via a receptor of the novel coronavirus, angiotensin-converting enzyme 2 (ACE2) [18]. As a well-characterized regulator of pulmonary vasodilation, ACE2may also exert beneficial secondary vascular effects such as dilation of lung vessels [19].

A strong association between β-thalassemia heterozygosity and increased mortality from COVID-19 was also observed. While in line with the rationale presented in our introductory remarks regarding hypoxemia and viral stress, this observation is in stark contrast to the protection of β-thalassemia heterozygotes against mortality due to COVID-19, which was reported at the beginning of pandemic [20,21]. 

In order to explain the difference between studies, we compared the days of hospitalization due to COVID-19 between carriers and non-carriers of the β-thalassemia trait (Figure 4). Significantly longer COVID-19 hospitalizations were observed for β-thalassemia trait carriers compared to non-carriers (*p* = 0.014). This observation suggests different carrier status-dependent dynamics between confirmed high risk non-carriers and no risk carriers of the β-thalassemia trait, during the early and later phase of the COVID-19 outbreak. According to this hypothesis, lagging end fatal episodes of COVID-19 positive β-thalassemia heterozygotes may yield reduced mortality during the early stage, which will be normalized during the later stage of pandemic. Analogous regional differences between prevalence of the β-thalassemia trait and mortality from COVID-19 due to the timing of the viral outbreak may also be observed. Such alleged differences may be further magnified by the short duration of data collection from participants at the onset of the pandemic. Thus, timing of data collection and study design differences may account for contrasting conclusions between studies. 

In light of the recent recognition of COVID-19 as a vascular disease leading to mortality from cardiovascular (CVS) failure [22], an operational overview of our observations regarding pathogenesis and treatment of β-thalassemia trait carriers is pertinent. CVS control is exerted through classical RAS inducing vasoconstriction via renin processed ANGII vasoconstrictors and counterbalancing non-classical RAS inducing vasodilation via ACE2 converted ANG1-7 vasodilators [23]. SARS-CoV-2 inhibits ACE2 expression and deranges CVS homeostasis [24]. Current strategies for COVID-19 treatment aim to suppress SARS-CoV-2 main protease activity, required to release active viral protein products and induce ACE2 expression [25]. Conversely, hyperlipidemia is a major systemic risk factor of CVS failure [26]. Ideally, effective treatment of CVS failure in COVID-19 patients should face a combination of viral and systemic risks. Statins appear to be ideal candidates for such combined treatment needs. On the viral front, statins bind to main protease and induce ACE2 expression, while on the systemic front they confer a potent lipid-lowering and anti-inflammatory effect [27]. 

Asymptomatic or mildly anemic β-thalassemia heterozygotes are in a state of threatened homeostasis, that if deranged may collectively induce expression of innate immune receptors CD45, Toll-like receptor 4, and CD32, reduce the ability to produce oxidative burst, and elevate membrane lipid peroxidation [28,29]. The compromised nature of response to stress inherent to β-thalassemia heterozygotes may explain the low threshold of COVID-19 symptoms required to begin treatment, which appear with considerable time lag and require considerably longer periods of hospitalization and ICU care. According to this comparison, also indicative to patients’ borderline resistance to stress, carriers of the β-thalassemia trait have a considerably higher risk of dying from COVID-19 compared to non-carriers (OR = 2.79).

Considering that COVID-19 induces CVS failure in this patient group and in view of the high mortality of β-thalassemia heterozygotes from COVID-19, the prescription of statins may improve this group’s clinical outcome. Thus, statin administration to COVID-19 positive β-thalassemia heterozygotes merits thorough consideration as a potential first line treatment option. Statins could represent an interesting treatment but, ideally, dedicated studies are needed.

## 5. Conclusions

In conclusion, carriers of the β-thalassemia trait are highly susceptible to COVID-19 and their early identification, consultation and treatment should be considered as a mandatory clinical practice, with important socioeconomic impacts for containment of the COVID-19 pandemic. Targeted studies from different study groups will be needed to assess the broader strength of our conclusions.

## Figures and Tables

**Figure 1 jcm-10-03645-f001:**
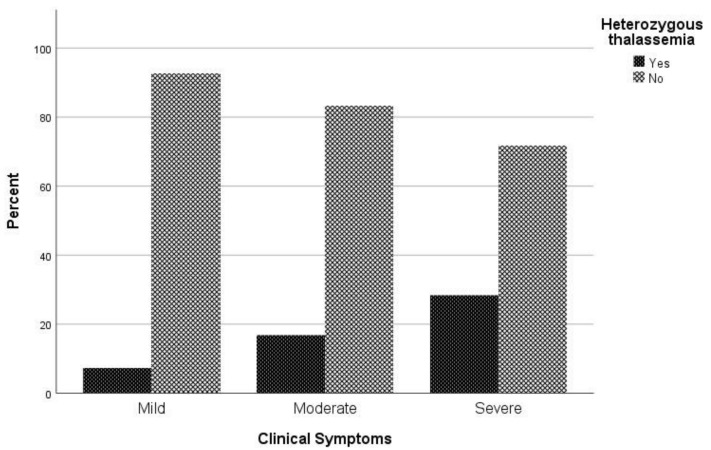
Proportion of β-thalassemia heterozygotes relative to non-carriers regarding clinical symptoms to COVID-19.

**Figure 2 jcm-10-03645-f002:**
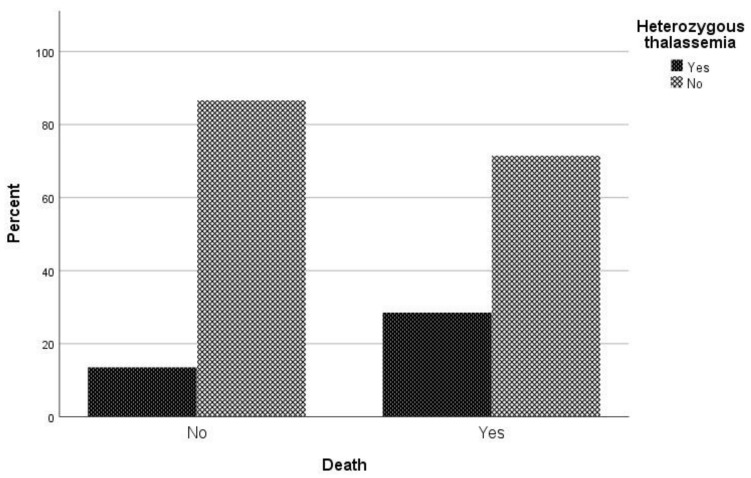
Proportion of β-thalassemia heterozygotes relative to non-carriers regarding mortality due to COVID-19.

**Figure 3 jcm-10-03645-f003:**
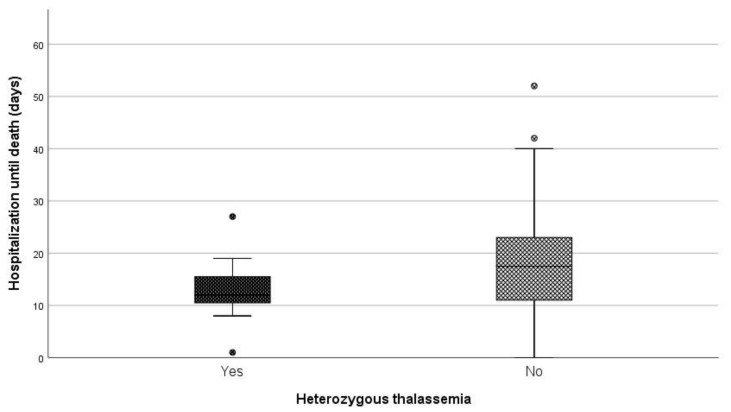
Days of hospitalization until death between carries and non-carriers.

**Figure 4 jcm-10-03645-f004:**
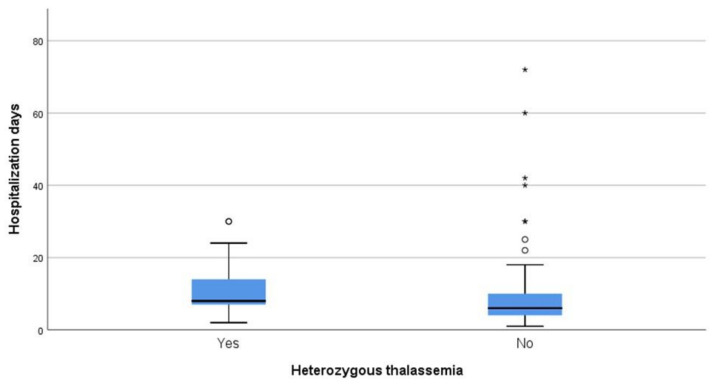
Days of hospitalization between carries and non-carriers that survived.

**Table 1 jcm-10-03645-t001:** Characteristics and COVID-19 clinical spectrum.

	Severity			Univariate	Multivariate Ordinal Logistic Regression (Severe and Critical vs. Others)	
	Mild (%)	Moderate (%)	Severe and Critical (%)	*p*-Value	*p*-Value	aOR with 95% CI
Sex (M/F)	34/34	67/46	52/22	**0.047** *	**0.023**	1.81 (1.09–3.01)
Age (median, IQR)	51.5 (34)	64.0 (17)	70.5 (15)	<**0.001** ±	<**0.001**	1.06 (1.04–1.08)
Atrial Fibrillation	17 (25.0)	32 (28.3)	33 (44.6)	**0.022** *	0.787	0.92 (0.49–1.71)
Respiratory Disease	5 (7.4)	13 (11.5)	14 (18.9)	0.104 *	0.325	1.47 (0.68–3.15)
Coronary Disease	7 (10.3)	23 (20.4)	20 (27.0)	**0.041** *	0.955	1.02 (0.50–2.09)
Diabetes	10 (14.7)	25 (22.1)	18 (24.3)	0.331 *	0.619	0.85 (0.45–1.60)
Neoplasia	7 (10.3)	11 (9.7)	11 (14.9)	0.529 *	0.209	0.61 (0.28–1.32)
Hyperlipidemia	21(30.9)	60 (53.1)	32 (43.2)	**0.014** *	0.138	0.65 (0.37–1.15)
Hypertension	24 (35.3)	62 (54.9)	56 (75.7)	<**0.001** *	0.104	1.67 (0.90–3.08)
β-Thalassemia Heterozygotes	5 (7.4)	19 (16.8)	21 (28.4)	**0.004** *	**0.002**	2.89 (1.49–5.62)

* Chi-square test, ± Mann–Whitney test; Bold is for the statistically significant results (*p*-value < 0.05).

**Table 2 jcm-10-03645-t002:** Characteristics and mortality due to COVID-19.

	Mortality		Univariate			MultivariateBinary Logistic Regression	
	Yes (%)	No (%)	*p*-Value	OR with 95% CI	RR with 95% CI	*p*-Value	aOR with 95% CI
Sex (M/F)	50/20	103/82	**0.022** *	1.99 (1.10–3.61)	1.67 (1.06–2.64)	**0.036**	2.09 (1.05–4.18)
Age (median, IQR)	72.5 (15)	61.0 (24)	<**0.001** ±	-	-	<**0.001**	1.06 (1.03–1.09)
Atrial Fibrillation	33 (47.1)	49 (26.5)	**0.002** *	2.48 (1.40–4.39)	1.88 (1.28–2.78)	0.201	1.64 (0.77–3.48)
Respiratory Disease	14 (20.0)	18 (9.7)	**0.027** *	2.32 (1.08–4.97)	1.74 (1.11–2.74)	0.297	1.61 (0.66–3.95)
Coronary Disease	20 (28.6)	30 (16.2)	**0.027** *	2.07 (1.08–3.96)	1.64 (1.08–2.49)	0.808	0.90 (0.39–2.09)
Diabetes	18 (25.7)	35 (18.9)	0.233 *	1.48 (0.77–2.84)	1.32 (0.85-2.05)	0.758	0.87 (0.41–1.91)
Neoplasia	10 (14.3)	19 (10.3)	0.367 *	1.46 (0.64-3.31)	1.30 (0.75–2.24)	0.395	0.67 (0.26–1.70)
Hyperlipidemia	30 (42.9)	83 (44.9)	0.773 *	0.92 (0.53–1.61)	0.94 (0.63–1.41)	**0.008**	0.38 (0.19–0.78)
Hypertension	52 (74.3)	90 (48.6)	<**0.001** *	3.05 (1.66–6.60)	2.30 (1.43–3.70)	0.198	1.67 (0.77–3.62)
β-Thalassemia Heterozygotes	20 (28.6)	25 (13.5)	**0.005** *	2.56 (1.31–4.99)	1.87 (1.24–2.80)	**0.010**	2.79 (1.28–6.09)

* Chi-square test, ± Mann–Whitney test; Bold is for the statistically significant results (*p*-value < 0.05).

**Table 3 jcm-10-03645-t003:** Characteristics and ICU admission due to COVID-19.

	ICU		Univariate			MultivariateBinary Logistic Regression	
	Yes (%)	No (%)	*p*-Value	OR with 95% CI	RR with 95% CI	*p*-Value	aOR with 95% CI
Sex (M/F)	36/17	117/85	0.186 *	1.54 (0.81–2.92)	1.41 (0.84–2.37)	0.305	1.45 (0.72–2.93)
Age (median, IQR)	66.2 (17)	60.4 (24)	**0.030** ±	-	-	0.649	1.01 (0.98–1.04)
Atrial Fibrillation	21 (36.9)	61 (30.2)	0.191*	1.52 (0.81–2.84)	1.39 (0.85–2.25)	0.966	0.98 (0.43–2.23)
Respiratory Disease	11 (20.8)	21 (10.4)	**0.043** *	2.26 (1.01–5.04)	1.83 (1.05–3.17)	0.205	1.80 (0.73–4.46)
Coronary Disease	16 (30.2)	34 (16.8)	**0.029** *	2.14 (1.07–4.27)	1.77 (1.08–2.92)	0.393	1.48 (0.61–3.59)
Diabetes	10 (18.9)	43 (21.3)	0.699 *	0.86 (0.40–1.85)	0.87 (0.48–1.64)	0.098	0.49 (0.21–1.14)
Neoplasia	4 (7.5)	25 (12.4)	0.466 ^†^	0.58 (0.19–1.74)	0.64 (0.25–1.63)	0.102	0.37 (0.11–1.22)
Hyperlipidemia	22 (41.5)	91 (45.0)	0.644 *	0.87 (0.47–1.60)	0.89 (0.55–1.45)	**0.033**	0.44 (0.21–0.94)
Hypertension	42 (79.2)	100 (49.5)	<**0.001** *	3.90 (1.90–7.99)	3.04 (1.64–5.63)	**0.001**	5.12 (2.04–12.87)
β-Thalassemia Heterozygotes	11 (20.8)	34 (16.8)	0.505 *	1.29 (0.61–2.77)	1.22 (0.68–2.18)	0.508	1.33 (0.57–3.06)

* Chi-square test, ± Mann–Whitney test, † Fisher’s exact test; Bold is for the statistically significant results (*p*-value < 0.05).

## Data Availability

Data are available on reasonable request.

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
