# Peer review of "Susceptibility of β-Thalassemia Heterozygotes to COVID-19"

_jcm, 2021, doi:10.3390/jcm10163645_

Round 1

Reviewer 1 Report

The authors have addressed the issues that I raised and is now in my view suitable for publication

Reviewer 2 Report

Authors have adequately modified the title and content of the discussion according to the reviewers' comments and according to the actual possibilities given by the study design.

This manuscript is a resubmission of an earlier submission. The following is a list of the peer review reports and author responses from that submission.

Round 1

Reviewer 1 Report

It is a well written paper and the main conclusion is that b-thalassemia heterozygotes is a high-risk group for COVID-19. I think that the design of the study does not support this "strong"conclusion, because there is a problem of bias. In Central Greece and specifically the in the prefecture of Larissa there is a high percentage of heterozygosity for b-thalassemia (10-15%,reference in https://repository.kallipos.gr/bitstream/11419/3091/1/Chapter_04_Loukopoulos.pdf, page 68). Apart from that, the number of the included patients is not sufficient in order to draw a safe conclusion and support the main conclusion of the study.

Reviewer 2 Report

This is an interesting paper describing a small cohort of COVID-19 patients and the influence of comorbidities on mortality. A negative association to mortality was found for hyperlipidemia and a positive one for beta-thalassemia trait status. The paper is well written, methodology and statistics are sound, references adequate. Conclusions are justified, whereas additional issues could be added in the argumentation to consolidate the results.

Although a good mechanistic explanation for the possible protective effect of hyperlipidemia status against mortality due to COVID-19 is offered (probably due to the pleiotropic actions of statins), argumentation on the thalassemia issue remains somehow uncertain. Following issues merit further discussion:

  1. Within the thalassemia cohort, mortality was equally distributed but in relation to the total n-numbers a higher OR was found for mortality. We cannot attribute this to thalassemia directly as a causal effect. It might be that by chance thalassemia patients who died had more comorbidities than the others. It would be interesting to have a descriptive table for comorbidities only for the thalassemia patients, to see whether they differed from the others.
  2. How was beta-thalassemia itself characterized, were all cases single gene traits or was there any compound heterozygosity with alpha-thal-traits?
  3. It is true that anemia can predispose to stress and in the context of ICU hospitalization anemia favors cardiac stress and death. What was the degree of anemia in the thalassemia cohort?
  4. Thalassemia trait patients normally tend to absorb more iron than needed from the gastrointestinal tract, in the absence of any kind of genetic hemochromatosis, and by the time they get a mild to moderate iron overload. This in turn could be a functional burden for the liver or the heart under additional stress or inflammatory conditions. Was iron overload characterized in the study cohort? Did thalassemia patients who died present any kind of hemophagocytosis syndrome/macrophage activation syndrome ?
  5. What was the cause of death in the thalassemia cohort, did it differ from the others?

Reviewer 3 Report

Dear authors,

Page 2

line 72

Current smoking status of participants was recorded in only 79 (31%) 72 study participants

This may represent a limit to your work, it would have been interesting to have a better idea of smoking history of the patients.

Page 8

line 230...

Considering that COVID-19 induces CVS failure in this patient group and in view of the high mortality of β-thalassemia heterozygotes from COVID-19, the prescription of statins may improve this group’s clinical outcome. Thus, statin administration to COVID- 19 positive β-thalassemia heterozygotes merits thorough consideration as a potential first treatment option.

This sentence describes an interesting idea, but it is only a hypothesis and a study should be ideally designed to demonstrate if it could be useful.

Maybe the sentence should end with something like: statins could represent an interesting treatment but ideally dedicated studies are needed.

For instance, any of the patients heterozygous for thalassemia were receiving statins?

Many open issues could be discussed in regard to your work, in my opinion the most important that should be clarified are:

-Did the patients undergo chest x ray and/or chest CT to evaluate COVID 19 related Pneumonia?

If yes, is there a difference between the patients heterozygous for thalassemia and the others?

-Causes of death/presumed causes of death in the patients of your population.

Are they available?

Is there a difference between the patients heterozygous for thalassemia and the others?

-statins, how may patients of your population were taking statins?

-hyperlipidemic patients, do you intend patients:

with hyperlipidemia?

patients treated for hyperlipidemia?

Both?

-number of patients enrolled should be compared to the population of the region were your hospital is based and to the COVID 19 epidemiology in your region.

This type of evaluation should also be compared to the prevalence of patients heterozygous for thalassemia in your region.

In fact although you performed univariate and multivariate analysis the small number of patients enrolled may be an important bias to your conclusions.

-since this may be the first article to consider b thalassemia heterozygotes at increased risk in COVID 19 are you sure that your statistics are well calculated?

Maybe a further statistic evaluation could be useful, for example a stepwise forward regression analysis.

-did you have any b thalassemia major patients with COVID 19 in your hospital?

-since the main topic is b thalassemia heterozygotes and COVID 19 the discussion should be focused more on this.

-a further analysis of b thalassemia heterozygotes patients could be useful, it would be interesting to know exactly what type of clinical characteristics they had (AF, hypertension...).

Kind regards

Reviewer 4 Report

This manuscript describes a retrospective analysis of a small group of patients hospitalised with PCR-positive COVID-19 diagnoses at a single tertiary hospital in central Greece during the latter part of 2020.  

The aim of the study was to determine whether being a beta-thalassemia carrier resulted in a higher susceptibility to COVID-19.  Other factors such as age, sex and certain co-morbidities were also investigated.

The study is well reported with Results neatly presented in three easy to read tables.  It is well-referenced and appropriately contextualised.

The authors report that their analysis showed that, as well as being older or male, there was an increased susceptibility to COVID-19 in beta-thalassemia heterozygotes.  Interestingly, this trait was not predictive for ICU admissions. 

The outcomes are statistically significant but how applicable they are to the larger population is not discussed.  The first sentence of the Discussion describes this analysis as a “pilot study”.   In my view this should be stated in the abstract and in the title.  As mentioned above, the study is at one hospital only and assessed data from just 255 patients.  During the study period there were approximately 120,000 PCR-confirmed cases of COVID-19 reported in Greece.  Whether the finding reported here is likely to be representative of the broader community would be worth knowing.  It would also be informative to readers if the predominant variant of SARS-CoV-2 that was prevalent in central Greece at the time of the study is noted in the manuscript.

In my view, this study should be  a pilot study from one hospital in one geographic location.  Such a report should then encourage others to determine whether there are similar correlations between beta-thalassemia heterozygosity and COVID-19 susceptibility in other COVID-19 patients.  Such larger analyses could then determine whether this is widespread across a county which has a higher proportion of this trait than most.  This could be particularly important in the context of treatment, noting that the authors quite appropriately discuss whether statin administration could be considered (final sentence in the Discussion).

Another minor point for consideration is whether there is a need for the four figures depicting the apparent influence of heterozygosity on the severity of COVID-19.  Some consolidation of these figures would seem appropriate since the data is described in the text.